# Histological Grade of Endometrioid Endometrial Cancer and Relapse Risk Can Be Predicted with Machine Learning from Gene Expression Data

**DOI:** 10.3390/cancers13174348

**Published:** 2021-08-27

**Authors:** Péter Gargya, Bálint László Bálint

**Affiliations:** Genomic Medicine and Bioinformatics Core Facility, Department of Biochemistry and Molecular Biology, Faculty of Medicine, University of Debrecen, Egyetem tér 1, 4032 Debrecen, Hungary; gargya.peter@gmail.com

**Keywords:** endometrium, biomarkers, endometrial cancer, machine learning, elastic-net, relapse-free survival, TCGA, RNA-seq, tumor grade, fertility preservation

## Abstract

**Simple Summary:**

Implementing machine learning methods into the RNA-seq data analysis pipelines can further improve the efficiency of data utilization in clinical decision making. In this article, we present how machine learning methods can be used to go one step further in data analysis of the global gene expression datasets, namely, to develop models that are able to classify individual cancer samples based on well characterized reference samples. We used the publicly available endometrial cancer sample RNA-seq datasets of the TCGA project to develop a model that can separate G1 and G3 cancer samples with an accuracy of 85%. Our model could also further stratify G2 samples into high-risk and low-risk subgroups. Moreover, with an iterative retraining approach, we could subselect twelve genes that performed similarly in the stratification. Our results were validated by the survival data of the patients.

**Abstract:**

The tumor grade of endometrioid endometrial cancer is used as an independent marker of prognosis and a key component in clinical decision making. It is reported that between grades 1 and 3, however, the intermediate grade 2 carries limited information; thus, patients with grade 2 tumors are at risk of both under- and overtreatment. We used RNA-sequencing data from the TCGA project and machine learning to develop a model which can correctly classify grade 1 and grade 3 samples. We used the trained model on grade 2 patients to subdivide them into low-risk and high-risk groups. With iterative retraining, we selected the most relevant 12 transcripts to build a simplified model without losing accuracy. Both models had a high AUC of 0.93. In both cases, there was a significant difference in the relapse-free survivals of the newly identified grade 2 subgroups. Both models could identify grade 2 patients that have a higher risk of relapse. Our approach overcomes the subjective components of the histological evaluation. The developed method can be automated to perform a prescreening of the samples before a final decision is made by pathologists. Our translational approach based on machine learning methods could allow for better therapeutic planning for grade 2 endometrial cancer patients.

## 1. Introduction

Endometrial carcinoma represents the fourth most frequent type of malignancy among women in developed countries [1]. Based on histological analysis, we differentiate between endometrioid and non-endometrioid types [2]. The tumor grade within the endometrioid category is determined based on the differentiation of the cells [3].

The tumor grade serves as an independent marker of survival [4] and may have an impact on therapy in line with the following: patients in the G3 group are considered to be high-risk patients and thus get adjuvant chemotherapy [5], while the G1 group may be suitable for fertility preservation therapy [6]. Patients of the G2 group in the middle of the scale fall short of this opportunity due to their worse prognosis, while at the same time chemotherapy is not recommended for them even though it would be justified for some patients. Thus, G2 patients are at risk of both under- and overtreatment. Additionally, histological evaluation is influenced by the non-negligible differences between the opinions of different pathologists [7,8,9]. Such factors are detrimental to the therapeutic decision-making process. Furthermore, as demonstrated by Visser et al. [10], the discordance between preoperative and postoperative tumor grades is most frequently observed in grade 2, thus a reliable, less subjective method could reinforce proper preoperative classification.

In recent years a growing number of approaches have emerged for the better characterization of cancers, which are based on the molecular profiling of tumors. The Cancer Genome Atlas (TCGA) project is one such outstanding example that performed detailed molecular profiling of tumors from copy number variation and mutation analysis, through RNA-Seq based gene expression and miRNA profiles. A direct result of the TCGA project is the identification of four molecular prognostic subgroups within endometrial tumors [11]. This comprehensive research project has revealed that in terms of their prognostic value molecular subgroups are superior to morphological classification, such as histological type or tumor grade [12]. The ongoing clinical trials rely on this new TCGA classification [13,14], but the gene expression data and results published by TCGA also carry in themselves the opportunity for the development of a procedure that could identify high-risk patients and enable better therapeutic planning. By using gene expression measurements, as primary data, certain research groups were able to estimate the prognosis of the disease [15] or they found promising biomarker candidates that were associated with recurrent early-stage endometrial carcinoma [16]. Others who used only clinical features failed to produce a highly accurate model [17]. Huang et al. used an RNA-seq based method to determine the presence of lymph node metastases [18]. A similar change of approach is also discernible in the case of cancer in other organs. In breast cancer, a tumor grade decision model was developed successfully, which was able to identify high-risk G2 patients [19]. In case of endometrial cancer previous research focused on predicting relapse, which can be influenced by various factors (e.g., surgical margin, therapy, etc.). Our main interest is to make the already used pathological classification more accurate by using RNA-seq as primary data -and machine learning as a tool to develop a robust method that can be translated into clinical practice. According to our knowledge, this approach has not been used in any previously reported study in the case of endometrial cancer.

Our objective was to divide G2 patients into high-risk and low-risk subgroups based on their gene expression profiles. Such a gene expression-based stratification approach for endometrial cancer could complement the current histological classification with an objective, semi-automated component that could reduce uncertainties in the therapeutic decision-making for such patients. In order to exploit the full benefit of global gene expression datasets, we decided to complement the gene expression data analysis methods with machine-learning classification tools. Our approach is founded on the objective laboratory procedure of RNA sequencing [20]. Our method could distinguish between G1 and G3 patients with high accuracy and in a fully reproducible manner, without having to rely on subjective components. Using the developed model, we could also stratify the G2 group into low and high-risk subgroups. The validity of this stratification was demonstrated by the patients’ survival data. As the developed model had good sensitivity and specificity, we decided to create another model but one based on a reduced number of genes instead of the whole transcriptome. As a result, we were able to achieve similar sensitivity and specificity while using only 12 transcripts. This approach may be suitable for further clinical validation in order to develop a routine clinical decision-making support tool, based on the qPCR measurement of a handful of genes [21].

Recent developments in RNA-based diagnostics make the implementation of RNA-based screens feasible by RT-QPCR from as small biological sample as a needle biopsy provides. As demonstrated by Visser et al. [10], the best agreement between preoperative and postoperative diagnosis was seen in the case of hysteroscopic sampling and not during dilatation and curettage. This observation suggests that directed, small volume sampling could improve clinical decisions. Therefore, the identification of RNA-based focused biomarkers that can be measured from a very small sample could contribute to the separation of high-risk and low-risk endometrial cancer patients.

According to our knowledge, this is the first report that proposes a shortlist of gene expression-based biomarkers that could help to make the transition from a three-level grading system to a binary one. Such a development is in line with the objectives of the clinical community as proposed by Soslow et al. [22] and with the novel developments generated within the TCGA project where the three-level grading system was replaced with the “low-grade” and “high-grade” binary system [11]. While TCGA subgroups provide molecular insight into the molecular events of endometrial cancer, the TCGA subgroups will change the risk-stratification of endometrial endometrium cancer. The tumor grade is, however, most likely to stay; thus, a more accurate grade assessment would be useful.

The above mentioned challenges of our work and our contributions to the field may be summarized as follows:The current 3-tier grading system of endometrioid carcinoma is not ideal for patients included in the grade 2 group as the result could be either under- or overtreatment.An objective, reproducible gene expression based method could separate patients included currently in grade 2 class into low-risk and high risk subgroups.We have developped a machine learning classifier method that can stratify grade 2 patients in high-risk and low risk subgroups.For translational purposes we refined this method to perform the stratification of patients with simillar performance using as few as 12 biomarker genes.Based on recent developments in RNA-based diagnostics, translation into clinical practice of our proposed biomarkers could contribute to the development of a robust, binary grade classification system by which more reliable preoperative classification could be implemented.

## 2. Materials and Methods

### 2.1. Downloading of Data

During our work we used R 3.6.1 and Python 3.6.5 programming environment, relying on their basic functions, as well as specific packages and modules. The machine-learning model and the simplified model using the minimally required number of genes were developed in Python, while R was used for the remainder of the analysis steps. Our samples were derived from the TCGA endometrial carcinoma study [11]. Using the TCGAbiolink 2.12.6 package, we downloaded the level three RNA-sequencing data altogether totaling 588 samples. The presence of a possible batch-effect was examined with the help of the MD Anderson Cancer Center TCGA Batch Effect viewer software (https://bioinformatics.mdanderson.org/public-software/tcga-batch-effects/ accessed on 30 October 2020). The extent of the batch effect was not found to be significant (DSC < 0.5), thus eliminating worry about any disturbing impact. The clinical data of the 546 patients of the samples were downloaded from www.cbioportal.org (accessed on 28 October 2020).

### 2.2. Screening of Data

The samples that were inadequate for our purposes based on clinical parameters were screened using the dplyr 1.0.3 package. Based on our applied conditions, we continued working with only those patients who had not received neoadjuvant therapy prior to the sample collection as neoadjuvant therapy could influence the gene expression profile [23]. A requirement for analysis was that their tumor grade classification was known and that they had a histologically verified endometrioid-type tumor. Of the remaining 406 people, the G2 patients were separated and the G1 and G3 patients were managed together.

### 2.3. Processing of RNA-Sequencing Data

The RNA-sequencing data studying 60,488 transcripts per sample were processed using the DESeq2 1.24.0 package [24,25]. (Other auxiliary packages: SummarizedExperiment 1.14.1, DelayedArray 0.10.0, BiocParallel 1.18.1, matrixStats 0.56.0, Biobase 2.45.0, GenomicRanges 1.36.1, GenomeInfoDb 1.20.0, IRanges 2.18.3, S4Vectors 0.22.1, BiocGenerics 0.31.5). As the first step in this processing, we removed those transcripts that had on average fewer than 4 detected reads for G1 and G3 samples overall. Thus 24,349 examined transcripts were left. We then normalized the gene expression matrix of the merged G1 and G3 groups with the help of the varianceStabilizingTransformation() function. We normalized the G2 group the same way based on the parameters received for the previous group.

We completed the principal component analysis with the plotPCA() function on the normalized values, using the top 500 transcripts, which showed the largest variance within the merged G1–G3 data set.

### 2.4. Machine-Learning Model Development

RNA-sequencing data consists of several examined genes; thus, it suffers greatly from the curse of dimensionality. To overcome this issue, we decided to use a regularized model. Elastic Net logistic regression offers an ideal solution for the development of a multidimensional, supervised classification model based on RNA sequencing between two groups, which uses L1 penalty and L2 penalty simultaneously, this way increasing bias but decreasing variance [26,27]. The L1/L2 ratio was regulated with alpha, while the scale of regularization with lambda hyperparameters. Regularized logistic regression can be easily implemented in a clinical setup, and it was proven to be effective for the analysis of RNA-seq data with a known tumor grade [19].

Our machine-learning algorithm using logistic regression based on Elastic Net regularization was created with the Python 3.6.5 programming language, with the help of the sklearn 0.23.2 module [28]. We divided our total data set made up by the G1 and G3 groups randomly in an 80:20 proportion to train and test sets. Within the training set, we performed nested cross-validation, with its outer loop made up by Monte-Carlo cross-validation (100 rounds) also in an 80:20 proportion. Within the training set of such a division, in the inner loop of the nested cross-validation, we looked for the optimal hyperparameters with 5-fold cross-validation. In the inner loop, we used the LogisticRegressionCV() function for model development and hyperparameter setting. We tested ten alpha values (l1_ratios = [0.001, 0.005, 0.01, 0.05, 0.1, 0.2, 0.3, 0.5, 0.7, 0.9]) and 10 lambda values (Cs = 10) chosen from the logarithmic scale between 10^−4^ and 10^4^ during hyperparameter tuning.

For the assessment of the performance of the models, we always used the area under the ROC curve drawn by the sensitivity and the 1-specificity as a reference. Other settings included following: fit_intercept = True, solver = “saga”, max_iter = 20,000.

Based on the ROC curve created in the process of training, we defined a decision threshold, which had the highest accuracy associated with it.

### 2.5. Analysis of Relapse-Free Survival

In cases where the model performs well in separating the G1 and G3 groups, with the trained model, we could make predictions on the missed G2 model, dividing them into a G3-type, high-risk G2, or a G1-type, low-risk G2 group based on the previously defined threshold value. Using the survival 3.1–11 and survminer 0.4.8 packages in R, with the help of a Cox proportional risk model and a Cox-Mantel test we compared the difference between the relapse-free survival of low-risk G2 and high-risk G2 groups.

### 2.6. Model Development with a Reduced Number of Genes

Using iterative re-analysis, we checked the minimum quantity of genes that need to be examined in order to achieve an accuracy that is almost identical to that of the full model. For this purpose, we arranged the genes in descending order based on the absolute value of their average coefficients in the outer loop, then with the help of a for-loop, we repeated the above-described training process. First only for the most important, then the first two most important genes, all the way to the top 200 most important genes. When deciding on the minimally required gene number, besides the AUC values we also considered their suitability for examination with the qPCR method. This means that, where possible, multiples of 12 were preferred.

With the help of the model with a reduced number of genes, we also subdivided the G2 patients into low and high-risk subgroups, and then following the same procedure, we completed the relapse-free survival analysis. The examination of the role of the most important genes was selected by consulting scientific publications.

## 3. Results

### 3.1. Patient Data Summary

In order to translate the gene expression datasets into a clinical decision support system, we investigated the datasets generated within the TCGA Project. Besides global gene expression datasets, the relevant clinical data of patients recruited within the project are also available. We identified 549 patient samples with endometrial cancer on the cBioportal.

We decided to exclude patients who had a non-endometrioid histological classification or received neoadjuvant therapy. Together with a dataset that could be considered a duplication, we defined the data sets of 139 patients as “Non-eligible”. As a result of this exclusion step, our investigation focused on the clinical datasets of 410 patients.

In order to combine the clinical data with the global gene expression datasets of the TCGA project, we screened the TCGA database for RNA-seq datasets of endometrium cancer patients and identifying 587. Aligning the 410 eligible patients and the 587 identified RNA-seq datasets, we found 406 patients who were eligible and had RNA-seq datasets available (Appendix A). These 406 patients had endometrioid-type cancer from a histological point of view; they did not receive neoadjuvant therapy prior to sample collection and had a known tumor grade value in the database. Of these 406 patients, 97 were classified as G1, 118 as G2, and 191 as G3 type. The mean age within these groups were 62.1 years for G1 (±10.4), 61.8 years for G2 (±11.6), and 63.3 years for G3 (±12.0) groups of patients. All clinical data of the 406 patients are presented in Appendix A. A detailed workflow of the data filtering steps is presented in Figure 1. Based on these, we could conclude that the TCGA database contained a relevant number of patients with endometrium cancer for whom both clinical data and global RNA-seq datasets were available and the patient numbers were relatively well equilibrated between G1, G2, and G3 subgroups without significant age differences between them.

### 3.2. The Need for Weighting between Genes

Our rationale for separating G2 patients into high-risk and low-risk subgroups was based on the presumption that the high-risk G2 patients would share similarities in their gene expression profiles with the patients in the G3 group, and the low-risk G2 patients would have a gene expression profile resembling the patients in the G1 group. Therefore, we first decided to develop a method to separate the G1 and G3 patients based on their global gene expression profile. By performing the classical gene expression normalization steps described in the Materials and Methods section, we used principal component analysis to reveal the largest differences between the G1 and G3 groups based on their gene expression profiles. As shown in Figure 2, the principal component analysis revealed that the G1 and G3 groups mostly overlap with each other based on gene expression, which justifies the need for weighing between different genes.

### 3.3. Histological Grade Can Be Predicted from Gene Expression

In order to develop a method to weigh the contribution of different genes to the G1/G3 classification, we decided to use a machine learning approach. For the development of a multidimensional, supervised classification model based on RNA sequencing between two groups, the Elastic Net logistic regression approach provided an ideal method. Our approach is presented in detail in Figure 3. We divided our total data set made up of the G1 and G3 groups randomly in an 80:20 proportion to training and test sets. Within the train set, we performed nested cross-validation, with its outer loop made up by Monte-Carlo cross-validation (100 rounds) also in an 80:20 proportion. Details of model development are described in Section 2.

During the training and checking of our model, the area under the ROC curve was 0.93 both with cross-validation and in the test by using 20% of the remaining data as seen in Figure 4, panels a and b. The best alpha was 0.50 and the best lambda was 0.04641589 during the hyperparameter setting. The sensitivity of the model on the test set was 0.87, its specificity was 0.74, its accuracy 0.85, positive predictive value 0.87 and its negative predictive value was 0.74. These values are presented in the confusion matrix in Figure 4, panel c. The threshold value for the ROC curve providing the best accuracy was 0.5159. The model was deemed adequately accurate for us to move on. Based on the quality control values of our model, the machine-learning-based approach could separate the G1 and G3 samples with high accuracy based solely on the gene expression profile of the samples.

### 3.4. Relapse-Free Survival Varies across G2-Subgroups

In order to separate G2 samples into low-risk and high-risk subgroups, we applied the model (developed in the previous stage) to the gene expression data of the G2 samples. Using the trained model, the G2 samples could be divided into low-risk and high-risk groups with 62 and 49 samples in the respective subgroups. In order to validate our approach, we compared the relapse-free survival (RFS) of the identified subgroups. As shown in Figure 5, there was a significant difference between the RFS of the two subgroups with the Cox-Mantel test (*p* = 0.037). The risk ratio of the Cox model was calculated at 2.54 (95% confidence interval: 1.025–6.307), therefore, our model was able to identify a major portion of high-risk patients from the perspective of relapse.

### 3.5. A Simplified Model Is Just as Accurate as of the Full Model

At this point, we were interested in whether a similar accuracy could be achieved while using a smaller number of genes than the 24,349 transcripts. For this purpose, we used the G1 and G3 datasets and arranged the transcripts in descending order based on their contribution to overall variability, represented by the absolute value of their average coefficients in the outer loop, and then repeated the former model development iteratively. Considering the average AUC values calculated by this method and the AUC values in the test set, we selected the 12 most important genes for our simplified model because it provides an easy-to-use number for a qPCR-based gene expression test (Figure 6, panel a and d).

Our simplified model with 12 genes could produce similarly good results. The area under the curve of the test set was 0.93 in this case as well (Figure 6, panel b). The best alpha was 0.001, while the best lambda was 0.006 during the hyperparameter setting. The sensitivity of the model on the test set was 0.92, its specificity was 0.68, its accuracy 0.84, positive predictive value 0.86 and its negative predictive value was 0.81. The threshold value for the ROC curve providing the best accuracy was 0.7421. These values are shown on the confusion matrix presented in Figure 6, panel c. As a result, we can state that using the machine-learning-based approach developed on the whole transcriptome datasets, we could narrow down the number of biomarkers to only 12 without reducing the performance of the model in separating between G1 and G3 samples based on their gene expression profiles.

In order to assess the performance of the simplified model based on the 12 biomarkers, we investigated samples with a G2 stage to separate them into high-risk and low-risk subgroups. Using the threshold value with the simplified model, the G2 samples were divided into low-risk and high-risk groups with 75 and 36 patients in them, respectively. To validate the clinical relevance of our procedure, we compared the relapse-free survival of the subgroups generated with only 12 biomarkers. There was a significant difference between the relapse-free survival of the two groups with the Cox-Mantel test (*p* = 0.0147, Figure 6, panel e). The risk ratio of the Cox model was calculated at 2.81 (95% confidence interval: 1.182–6.682), thus our simplified model was also able to identify the high-risk G2 patients. As a result, we succeeded in developing a method to stratify G2 patients into low-risk and high-risk subgroups based on gene expression data, moreover, we identified 12 selected biomarker genes that could be further evaluated in a clinical study.

We further validated each of these 12 genes individually by using https://kmplot.com, accessed on 19 August 2021) [29,30], an online tool that relies on the same TCGA cohort of 543 patients (Figure 7). The overall survival of patients was affected by 9 out of 10 genes (pseudogenes were not included). Based on these plots we concluded that our 12 genes not only predict relapse but have a long-term prognostic value as well.

## 4. Discussion

Our classification model used the machine-learning algorithm to successfully identify the gene expression profiles of G1 and G3 patients and the differences between the groups. Based on this, the trained model was able to identify the high-risk G2 patients, thus complementing the current histological classification. Our simplified model with 12 genes produced similarly good results. The latter may be suitable for further development as a qPCR-based clinical test that is significantly cheaper and faster than a global transcriptome approach. At the same time, RNA isolation and qPCR itself are processes that may be highly automated [20,21]. These attributes entail the possibility that the gene expression profiling of endometrial carcinoma may, in the long run, due to related financial and human resource costs, serve as an alternative to the procedure from the TCGA study using immunohistochemistry and POLE gene sequencing [13]. Moreover, in the case of an adequately large sample, the subgroups of the TCGA groups may also be described with the same method, which also justifies the use of technologies relying on gene expression as additional stratification tools.

Of the 12 genes used in the simplified model, some of the protein-coding genes are known to have a role in the pathogenesis of endometrial carcinoma, while for the time being other genes have been studied in detail but only in tumor types other than endometrial carcinoma. We found that high expression of FOXB1, HABP2, EDN3, B3GAT1-DT, and DKK4 was associated with low-risk phenotype, while high expression of RPL41P1, MAL, UCHL1, CRABP1, PEG10, RPS28P7, and MLF1 was responsible for more clinically aggressive behavior.

Similar to the other Dickkopf proteins, DKK4 controls the WNT signaling pathway [31]; its differing expression and role in tumor genesis and tumor progression have already been described in multiple tumors, including endometrial carcinoma. The MLF1 gene codes an oncoprotein that is responsible for the determination of the phenotype of hematopoietic cells [32]. Translocation between MLF1 and the nucleophosmin has been associated with myelodysplastic syndrome and acute myeloid leukemia. Its role in endometrial carcinoma is less clear. B3GAT1-DT is a lncRNA coding gene, its role has been studied only recently and was associated with diffuse large B-cell lymphoma [33]. B3GAT1-DT was also associated with spontaneous regression and neuronal differentiation in neuroblastomas [34]. Since miRNAs and other non-coding RNAs are described as important prognostic markers in endometrial cancer [35,36], we note that B3GAT1-DT could be also an important marker, worthy of further study. EDN3 is a player in the endothelin signaling pathway, which regulates cell differentiation, proliferation, and migration. Abnormalities caused by EDN3 were primarily described in breast cancer [37], as a result of which the survival and invasion ability of the cells changed. The HABP2 gene codes an extracellular protease, which plays a role primarily in coagulation and fibrinolysis but has also been mentioned in adenocarcinoma as an enhancer of tumor progression [38]. The PEG10 gene has been identified in various tumors, including that of the endometrium, as a gene that has a role in enhancing cell division [39]. CRABP1 is a retinoic-acid-binding protein, the increased expression of which has already been associated with bad prognosis and increased proliferation in endometrioid endometrial carcinoma [40]. The product of the UCHL1 gene, ubiquitin C-terminal hydrolase L1, has been described as a key player in the metastases formation of tumors in most tumor types, thus also in endometrial carcinoma [41]. FOXB1 is a DNA-binding transcription factor protein the specific role of which has not yet been examined in the tumorigenesis of the endometrium [40,42]. The protein coded by the MAL gene is a strongly hydrophobic integral membrane protein. It localizes in the endoplasmic reticulum of T-cells and participates in their signal transduction. It plays an additional role in the creation and maintenance of membrane microdomains rich in glycosphingolipids. The reduction in the expression of MAL has been associated with the emergence of different epithelial tumors [43]. Interestingly, the RPL41P1 and RPS28P7 pseudogenes were also among the top 12 most important transcripts in terms of classification. Based on publicly available resources aggregated in the UCSC Genome browser, we investigated the possibility for the expression of these two pseudogenes. As we hypothesized, both pseudogenes are indeed subject to transcription. RPS28P7 overlaps with the EST DA063139 [44]. Moreover, this region overlaps with the intronic region of mir4300HG, therefore this region might be transcribed. The RPL41P1 region overlaps with the piRNA DQ585299. This region is shown to be expressed in a large variety of cancer samples. Further studies are needed to map the functional importance of these transcripts [45].

Besides the development of the qPCR-based clinical decision support system, further study of these genes may be justified because they provide a great amount of useful information on the molecular characteristics of endometrial carcinoma, and the products of these genes may even serve as potential drug targets. As some non-coding RNA molecules can be identified from these regions, further studies are needed to gain a better understanding of these non-coding RNA molecules.

As a potential future development, the immunohistochemical investigation of our proposed biomarkers could be considered. Two of our biomarkers that are present on the RNA level in a variety of databases were earlier classified as pseudogenes; therefore a limited number of prior data is available. On the other hand, our proposed gene list can provide good performance (e.g., AUC higher than 0.8) even with three genes therefore immunohistochemical measurements should be considered at a later stage for at least some of the genes.

The goal of the TCGA project was to generate comprehensive molecular maps of a large variety of cancer types. In the project 33 cancer types were included [11,46]. The generated molecular profiles include DNA sequence, copy number, methylation, mRNA, and miRNA together with some dedicated protein profiles. More than 25,000 samples were processed in the project. The samples were shipped from collection sites to a Core Resource Center together with a pathology report and were reviewed by a board-certified pathologist in order to verify that the specimen was consistent with the diagnosis and to assess the QC parameters of the samples. As the main goal of the overall TCGA project was to generate the molecular profiles of the collected samples, no dedicated board of pathologists were involved in deciding the histological classification of the samples. To be noted, for some dedicated cancer types such as sarcomas, follow-up studies involved a dedicated board of pathologists for histological classification [47], but not for endometrium cancer [11]. This aspect of the sample collection might be considered a potential source of error and is a serious limitation of the dataset used in our study.

Another important aspect that should be considered if we investigate the potential limitations of our study is that the datasets used in our investigations, generated in the TCGA Project, were generated from samples that contained not only cancerous cells but very likely normal cells, necrotic cells, and all other cell types characteristic of cancer tissues. Therefore, the generated molecular maps do not show only the profiles of the cancer cells but the average of a variety of cells present in the processed bio-samples. While this is a limitation from a basic science point of view, it can be a benefit for a translational approach that aims to identify the most robust biomarkers present in a cancer sample.

Overall, our results and the additional research that can follow up our results could help to further improve the living prospects of women suffering from endometrioid endometrial carcinoma. By further stratifying grade 2 patients into high-risk ad low-risk subgroups our development can contribute to the implementation of a binary FIGO grading proposed by Soslow et al. [22] and better therapeutic decisions for endometrium cancer patients. Our proposed gene expression-based method suggests that besides the mutation-based tests the procedures examining gene expression also have a future in determining the prognosis of endometrial carcinomas.

## 5. Conclusions

Speeding up the translation of basic scientific results into clinical practice requires novel methods that can provide reliable pre-screening information for the pathologist. Global gene expression analysis performed with RNA-seq became technically available in every major medical center; still, these methods are not used routinely in medical decision making due to the complexity of the generated data. Implementing machine learning methods into the data analysis pipelines can speed up the final clinical decision-making and thereby, reduce the waiting times of the patients for a final diagnosis.

While machine learning methods have been implemented in automated image analysis and have significantly improved the quality of clinical decisions, we observe a delay in their implementation into nucleic acid based clinical decisions. In this article, we present a machine learning based method that can stratify the G2 endometrioid cancer patients into low-risk and high-risk subgroups based on their gene expression profiles. With iterative retraining, we have developed a model that relies on only a set of twelve genes and has 84% accuracy. Our approach could be used to generate semi-automated prescreening reports and therefore speed up the final clinical decision-making by the pathologist.

## Figures and Tables

**Figure 1 cancers-13-04348-f001:**
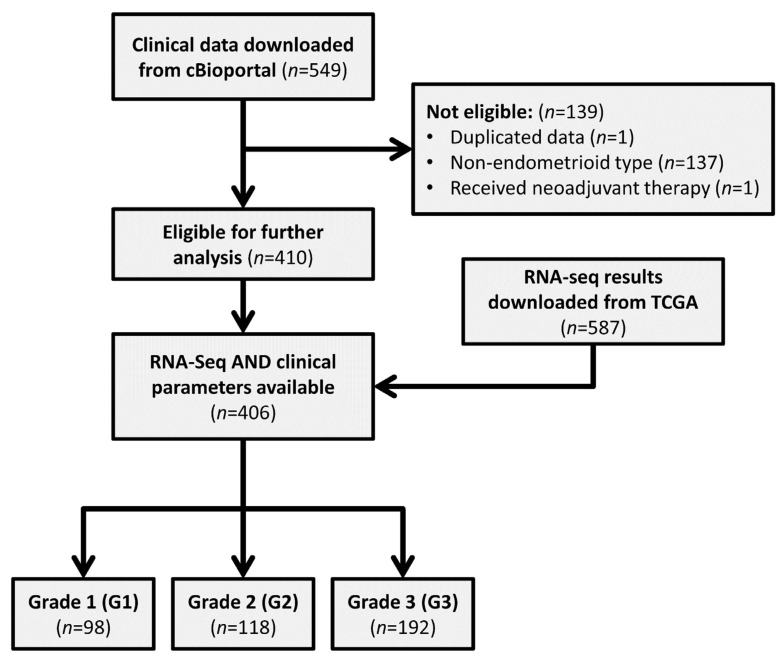
Workflow of sample filtering and processing.

**Figure 2 cancers-13-04348-f002:**
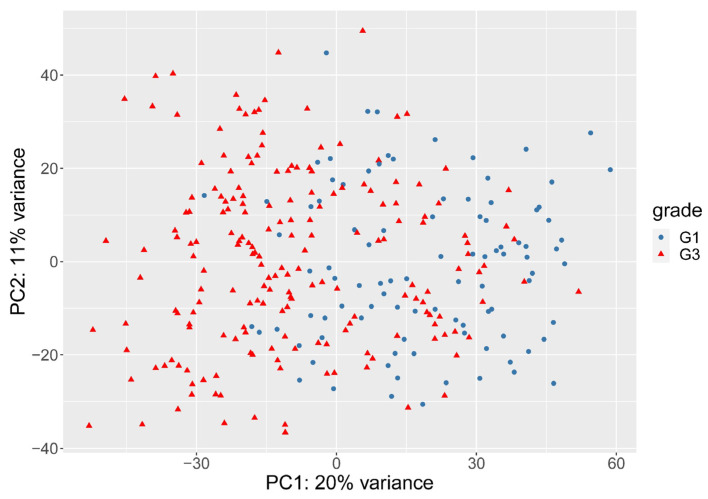
Principal component analysis of G1 (blue dots) and G3 (red triangles) groups.

**Figure 3 cancers-13-04348-f003:**
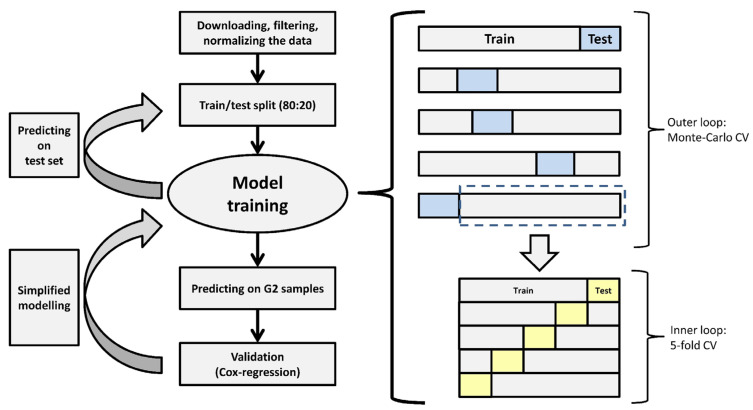
Flowchart of data processing and modeling. See the materials and methods section for details.

**Figure 4 cancers-13-04348-f004:**
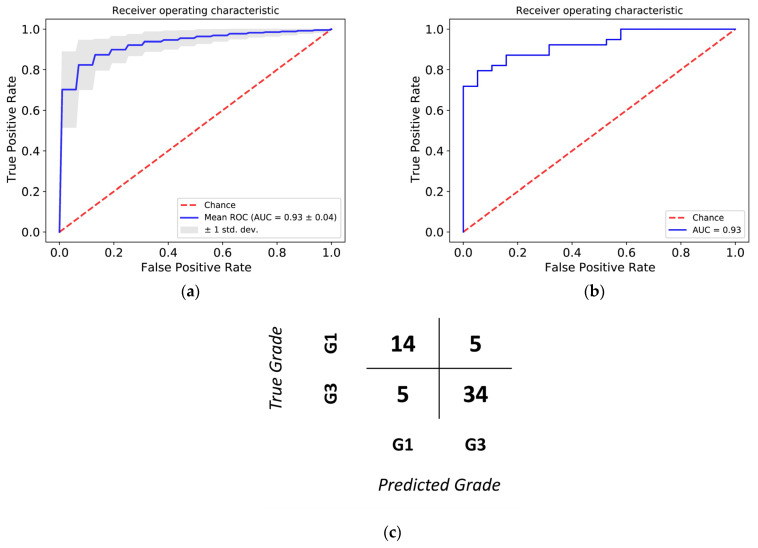
ROC curves of the cross-validation rounds (**a**) and the test data (**b**). The blue line represents the mean AUC value and the grey area represents the standard deviation. (**c**) Confusion matrix of the test data.

**Figure 5 cancers-13-04348-f005:**
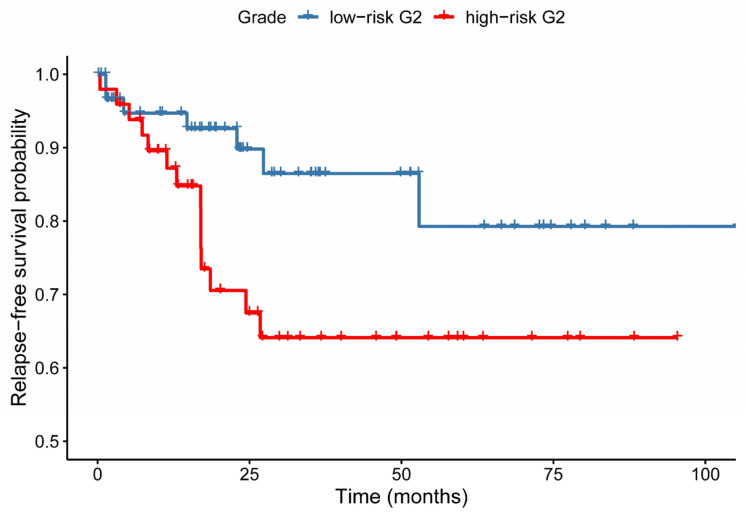
Kaplan–Meier curves of relapse-free survival between groups predicted by our machine learning model. Blue: low-risk G2, red: high-risk G2. Cox-Mantel test *p*-value = 0.037.

**Figure 6 cancers-13-04348-f006:**
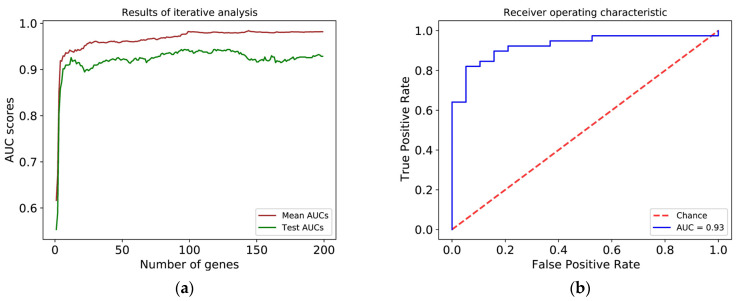
(**a**) Results of the iterative retraining during the search for the minimum number of eligible genes. The brown line represents the mean AUC value during cross-validation rounds; the green line represents the AUC value based on the test set. (**b**) ROC curves of the test data. (**c**) Confusion matrix of the test data. (**d**) Panel showing the top 12 most relevant genes’ ID and their elastic-net coefficient respectively. Genes with negative coefficients are related to the low-risk group, while genes with positive coefficients are related to the high-risk group. (**e**) Kaplan–Meier curves of relapse-free survival between groups predicted by our simplified machine learning model. Blue: low-risk G2, red: high-risk G2. Cox-Mantel test *p*-value = 0.0147.

**Figure 7 cancers-13-04348-f007:**
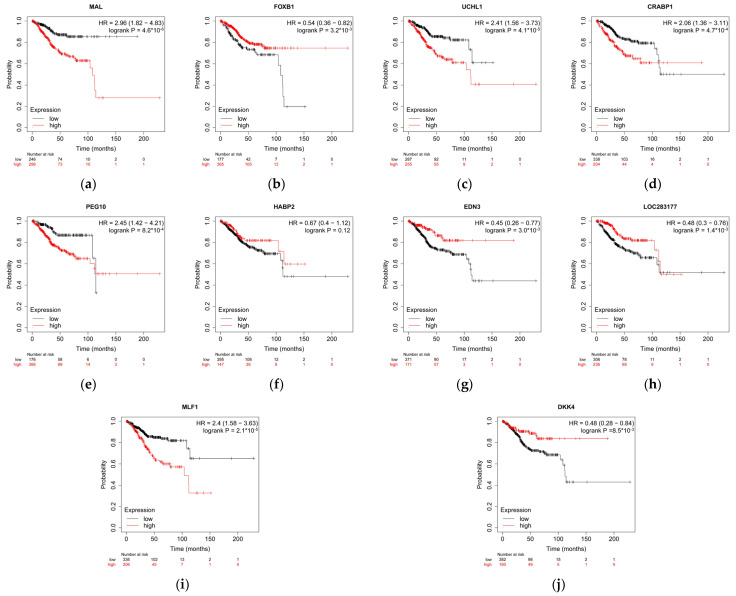
(**a**–**j**) Kaplan–Meier curves of overall survival between groups with high gene expression and low gene expression of selected genes. 9 out of 10 genes had passed the Cox-Mantel test with *p*-value < 0.05. We note that RPL41P1 and RPS28P7 annotated as pseudogenes could not be found in the KM-plotter database. Here, LOC283177 is used as an alias of B3GAT1-DT.

## Data Availability

Publicly available datasets were analyzed in this study. This data can be found here: https://portal.gdc.cancer.gov/ (accessed on 28 October 2020) and www.cbioportal.org (accessed on 28 October 2020). Codes used for the data analysis are available at: https://github.com/gargyapeter/ucec_ml_grade2021.git (accessed on 23 August 2020).

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
