# Peer review of "Histological Grade of Endometrioid Endometrial Cancer and Relapse Risk Can Be Predicted with Machine Learning from Gene Expression Data"

_cancers, 2021, doi:10.3390/cancers13174348_

Round 1
Reviewer 1 Report
In this paper entitled “Histological Grade of Endometrioid Endometrial Cancer and Relapse Risk Can Be Predicted With Machine Learning From Gene Expression Data”, authors developed machine learning based model which classify grade 1 and grade 3 samples using RNA-seq data. The authors have done an great job at describing the problem, the methods and the results but there are some key aspects that need to be addressed.
- Elaborate more on the novelty of the paper.
- What is the motivation behind employing machine learning methods, not deep learning which is the trend nowadays.
- Paper contributions and work challenges should be clear enough for readers. I would recommend making them as bullet points.
- The research gap of previous methods on the same topic has to be elaborated along with listing the previous methods. Authors need to do more research on this topic. Please see these recently published papers. (1) Predicting prognosis of endometrioid endometrial adenocarcinoma on the basis of gene expression and clinical features using Random Forest (PMID: 31423227), (2) The application of machine learning for predicting recurrence in patients with early-stage endometrial cancer: a pilot study (PMID: 33371658), and (3) Identification of early-stage recurrence endometrial cancer biomarkers using bioinformatics tools (PMID: 32705231).
- Author also mentioned the term ‘teaching set’ in the manuscript, does they mean about training set or testing set? Please make it clear otherwise it make confusion for the audience.
- Code link provided in the manuscript shows ‘This repository is empty’. The model classification code as well as training and test set should be made available online on a Github repository, otherwise it is not possible to reproduce the results shown in this manuscript.
Reviewer 2 Report
This article dealt with the exploration of biomarkers (prognostic markers) in grade 2 endometrioid endometrial cancer (EEC) using bio-informatics method of machine-learning methods.
As for data, authors used results of RNA-sequence from TCGA project. The methods used were machine-learning method with the data of grade 2 EEC by Monte-Carlo cross validation in 100 rounds. G1 EEC and G3 EEC were control in relapse free survivals to select transcripts and genes.
The analyses methods and the statistic procedures were correct and the results were rigorous in re-assurance by ROC analysis.
This article is interesting and informative for readers and worthy for publish if the following two sampling biases in macroscopically and microscopically.
questions about biases.
#1. Were the sampling specimens for pathological diagnoses appropriate? How was the kappa-value among pathologists in TCGA? As you know, GOG (now NRG-GOG) have pathological committee and nearly 20-25 pathologists check every specimen. The diagnosis of grade 2 is some-what ambiguous rather than G1 and G3. So, it is important and necessary to central pathology.
#2. I think that the sampling tumor cells for RNA-seq were performed in Laser Microdissection from formalin embedded slices not to be contaminated normal tissue, especially, stromal tissue cells.
How did the collection of tumor cells alone not using the crypt isolation method in glands (Nagasawa, Sugai et al. 2016), which method completely discriminate among tumor cell, normal endometrial cell, and stromal cell?
Ref.
Nagasawa, T., et al. (2016). "Molecular Analysis of Single Tumor Glands Using the Crypt Isolation Method in Endometrial Carcinomas." Int J Gynecol Cancer 26(9): 1658-1666.
OBJECTIVE: Endometrial adenocarcinomas are characterized by the presence of many single tumor glands in which multiple genetic changes have accumulated. To elucidate the differences in molecular abnormalities among single tumor glands, individual tumor glands were analyzed and microsatellite alterations (loss of heterozygosity (LOH) and microsatellite instability [MSI]) were examined using the crypt isolation method in glands from each tumor from patients with endometrial carcinoma. METHODS: Twenty-five patients with endometrial adenocarcinoma who underwent surgery were included in this study. We obtained cancerous individual isolated tumor glands from each patient using the crypt isolation method. For LOH and MSI analyses, we used 15 microsatellite markers (3p, 5q, 10q, 13q, 17p, 18q, BAT25, and BAT26) and the promoter regions of 6 genes (transforming growth factor beta receptor II, BAX, insulin-like growth factor II receptor, E2F4, MutS homolog 3, and MSH6). RESULTS: Loss of heterozygosity was detected in 8 (32%) of 25 patients, and MSI was detected in 9 (36%) of 25 patients. Some MSI-positive carcinomas had LOH in single tumor gland samples, and the coexistence of LOH and MSI was confirmed. In 16 (64%) of 25 cases, intra-tumoral genetic heterogeneity among single tumor gland samples was detected. CONCLUSIONS: By analyzing multiple single tumor glands within the same tumor, we found that endometrial adenocarcinoma was composed of various tumor glands with different molecular abnormalities, even in a limited region within the same tumor.

Reviewer 3 Report
Very interesting paper, well-written with clear study design, innovative and clinically relevant.
There are some minor suggestions to improve the manucript.
Treatment in endometrial cancer is initially surgical, and whether or not lymph node surgery is needed is usually based on preoperative tumor grade. As demonstrated by Visser NC et al. (Obstetrics & Gynecology 2017) the discordancy between preoperative and postoperative tumor grade is most frequent obeserved in grade 2 and therefore reinforces proper classification already preoperative. This issue might be added in the introduction / background and clinical context.
Although TCGA is an important prognosticator, with worst outcome in the CNH (abn p53), still can at least be partially explained by the fact that about 50% of CNH have ae advanced stage disease. It would be worthwhile to know if the set of 12 most relevant transcripts were related to tumor stage, and as such could indicate the need for surgical staging procedure if 'high risk' profile is identified.
Were the transcripts as identified related to the TCGA groups ? As still about 70% of EC patients can be attributed to MSI and NSMP with intermediate outcome, these transcripts might be interesting for refining TCGA subgroups.
With respect to clinical implementation, transcripts might be evaluated by immunohistochemistry, the authors should speculate on this in the discussion, as well as implementing in preoperative setting.
Based on the Kandoth paper (2013) the grade 1-3 tumors were classified 2 in relation to TCGA might be
Round 2
Reviewer 1 Report
Authors address all the questions and concerns.
This manuscript is a resubmission of an earlier submission. The following is a list of the peer review reports and author responses from that submission.